# The Effect of the Cultivar and Harvest Term on the Yield and Nutritional Value of Rhubarb Juice

**DOI:** 10.3390/plants10061244

**Published:** 2021-06-18

**Authors:** Ivana Mezeyová, Ján Mezey, Alena Andrejiová

**Affiliations:** 1Department of Vegetable Production, Faculty of Horticulture and Landscape Engineering, Slovak University of Agriculture, Dražovská 4, 94901 Nitra, Slovakia; alena.andrejiova@uniag.sk; 2Department of Fruit Growing, Viticulture and Enology, Faculty of Horticulture and Landscape Engineering, Slovak University of Agriculture, Dražovská 4, 94901 Nitra, Slovakia; jan.mezey@uniag.sk

**Keywords:** rhubarb, sugars, organic acids, juices, yield of petioles

## Abstract

Since scientific interest in rhubarb from a culinary point of view is a relatively new issue, the aim of this study was to test five edible cultivars of *Rheum rhabarbarum* L. (‘Poncho’, ‘Canadian Red’, ‘Valentine’, ‘Red Champagne’, and ‘Victoria’) from a specific culinary perspective, i.e., processing into juice. Total yields (t/ha) were established in six harvests during a two-year field experiment. For juice production and subsequent laboratory analysis, rhubarb petioles from two different harvest terms were used (i.e., harvest term A (HTA) and harvest term B (HTB)). Analyses of total sugar, glucose, fructose, total soluble solids (TSS), total acidity, malic acid, and pH level were determined by FT-IR spectrophotometer. Total yields of petioles varied between 28.77 t/ha (‘Canadian Red’) and 45.58 t/ha (‘Red Champagne’) at a density of 11,000 pl/ha. ‘Red Champagne’ significantly (*p* < 0.05) reached the highest juice yield potential (85%) and the highest values of glucose (9.97 g/L), total soluble solids (4.37 g/L), and total sugars (54.96 g/L).

## 1. Introduction

Rhubarb (*Rheum rhabarbarum* L.) is a perennial plant from the family Polygonacea with valuable nutritional and medicinal properties [1] and high potential for cultivation as it provides early yields when the vegetable supply to market is deficient. Rhubarb is grown for its large, thick leaf stalks or petioles that are consumed as food [2]. The leaf stalks are of various widths, with a range of 30–50 mm in diameter. The stems range in length from 300 to 500 mm and can be green to anthocyanin in color. The position of the stems can be upright or semi-upright to horizontal [3]. The enlarged petioles develop from a central crown of the rhubarb plant. Petiole color is associated with rhubarb quality and the order of preference is red, pink, and green [2]. Although rhubarb is a well-known vegetable, scientific interest in this plant is a relatively new issue; most of the evidence of its biological activities and therapeutic potential derives from the last 15 years [4]. It shows high levels of both polyphenol content and antioxidant capacity in edible parts [5], which are petioles characterized by very high antioxidant properties and rich in many compounds that have a pro-health effect on the human body [6]. There is a wide variety of rhubarb cultivars that contain bioactive compounds such as flavonoids, anthraquinone, glycosides, tannins, volatile oils, and saponins [1]. The chemical composition of rhubarb includes anthraquinones, anthrones, stilbenes, tannins, polysaccharides, etc., which show extensive pharmacological activities including gastrointestinal regulation, anticancer and antimicrobial properties, hepatoprotection, cardiovascular and cerebrovascular anti-inflammatory protection, etc. [7,8]. The main characteristics of rhubarb quality are taste and aroma, which depend on the chemical composition [9]. Studies on the phytochemical composition of different species of rhubarb have provided information on the presence of a variety of inorganic and organic acids (including tartaric, oxalic, citric, malic, and ascorbic acids) [4].

Increased interest in the application of natural biologically active substances in human nutrition has led scientists to develop functional foods. The production of functional drinks is an ever-changing aspect of the beverage industry, as there are still new and prospective trends in this area [10]. Some examples of these kinds of foods include juices that are not obtained from concentrate (NFC) and freshly squeezed non-pasteurized juices (FS). These juices are obtained from the fruit tissue by pressing and centrifugation of the pulp [11]. Nutritionists recommend the consumption of fruit or vegetable juices instead of replacing natural products with synthetic vitamin and mineral supplements [12]. Fruit juices as functional drinks offer promoting good health by reducing the risk of serious illness. These beverages contain ingredients that provide specific benefits and are enriched with vitamins, minerals, amino acids, fiber, or antioxidants [13]. Products made from rhubarb have a favorable taste due to a high content of organic acids and rhubarb stalks taste best in early spring when they are ripe. Because rhubarb leaves are toxic due to their content of oxalic acid [14], the petioles are used for processing juices.

It should be noted that the scientific literature lacks studies on this plant and its application in the food industry. Therefore, for constant development and consumers’ search for new solutions, it is necessary to focus attention on this plant [6]. It has been utilized for thousands of years for medicinal purposes, but only recently identified for its culinary use [15]; it was not until the 18th century that the culinary use of petioles was first reported [16]. In culinary rhubarb, the high oxalate content is a major drawback. A renewed interest in rhubarb production is now directed towards the use of stalks from low-oxalate cultivars as a cheap filler for industrial production of marmalade, jam, and syrup [17]. It has potential importance in the food and pharmaceutical industries to expand the range of products in the future by adding a new product with outstanding antioxidant properties [6]. Therefore, the aim of this study was to test selected rhubarb cultivars from a specific culinary perspective, i.e., processing into juice. From qualitative parameters, the sugars and acids were established in rhubarb juices which are important from a sensory analysis point of view. In addition, the juice yield of the plant was tested as an important parameter in the case of practical uses.

## 2. Materials and Methods

This 2-year vegetation field experiment was carried out at the Botanical Garden (BG), Slovak University of Agriculture (SUA), in 2018 and 2019 (Nitra, Slovak Republic, 48°18′ N, 18°05′ E, 144 masl).

### 2.1. Rhubarb Cultivars’ Characterization

Morphological characterization and quantitative parameters of petioles of selected cultivars of *Rheum rhabarbarum* L. are summarized in Table 1 and Table 2.

### 2.2. Soil and Climate Characteristics

The soil type of the experimental area is brown soil to chernozem on loess and loess loams and the part along the river Nitra belongs to the area of fluvial soils, where the original soil type was fluvial and fluvial glue. The plants were grown in growing substrate suitable for *Rheum rhabarbarum* L. In terms of climate classification of the region, Nitra is situated in a warm and dry area of Slovakia. The evaluations of experimental years according to climate normals are given in Table 3 and Table 4.

### 2.3. Organization of the Experiment

The cultivation of plant material was carried out in accordance with modern agrotechnical practices of rhubarb cultivation. The plants were purchased from the perennial nursery VICTORIA (Čab, Slovakia) and were planted on 5 May 2017, in spacing 1.0 × 0.9 m (density of 11,000 pl/ha). A total of 100 petioles from each cultivar were monitored. The petioles were harvested at the stage of consumption maturity at BBCH 39/49 (rosette development completed/typical leaf mass reached). When collecting the petioles, the minimum 20 cm length and 1 cm width (diameter) was respected. The harvest was conducted manually and gradually over two months (6 realized dates in the case of total yields and morphological descriptions) from the middle of April to the middle of June, at intervals of 1 time per week. For juice production and subsequent laboratory analysis, the rhubarb petioles from two different harvest terms were used, in a time span of 1 month, i.e., harvest term A (HTA) carried out on 18 May 2018 and harvest term B (HTB) on 17 June 2018. In 2019, HTA was carried out on May 13 and HTB was carried out on June 13.

### 2.4. Sample Preparation

Subsequent washing and analyses were conducted in the Laboratory of Beverages of Research Centre AgroBioTech SAU in Nitra. A total weight of 500 g of petioles was washed and harvested in 100 mm long pieces. They were processed on a commercial fruit juicer with 1200 rpm without heating (Magimix Duo Le Plus XL, Vincennes, France). The total volume of obtained juice was approximately 400 mL from each rhubarb cultivar replication. Immediately after juicing, the average juice yield from each cultivar was calculated. Following that, juice was strained through sieves of 0.2 mm diameter and processed through a centrifuge (Hettich BV-20, Tuttlingen, Germany) for 120 s at 6000 RPM in 8 × 10 mL plastic test tubes.

### 2.5. Juice Yield

The efficiency of juicing was calculated using the formula according to Wilczyński, 2019 [11]:Wj (%) = Mj/Mi × 100
where Wj is the efficiency of juicing (%), Mj is the mass of juice after juicing (kg), and Mi is the mass of input material (kg).

### 2.6. Testing of Selected Parameters in Juice

Analyses of total sugar, glucose, fructose, total soluble solids, total acidity, malic acid, and pH level were analyzed by FT-IR spectrophotometer (Alpha Wine Analyzer, module for juices, Bruker Optics, Billerica, MA, USA) [19] which uses invisible infrared light of which certain wave lengths are absorbed by the sample, depending on its characteristics. It analyzes the sample by utilizing the attenuated total reflection (ATR) measurement technique. The core of the ATR technique is a diamond crystal that reflects the incoming light at a right angle to the detector. A sample is placed on top of the diamond and a fraction of the light penetrates the sample and is attenuated according to the absorption characteristics of the sample. This attenuation is measured by the detector and transformed into a spectrum [20,21]. The parameters were determined in three replicates for each sample of extracted juice.

### 2.7. Statistical Analyses

The analysis of variance (ANOVA), the multifactor analysis of variance (ANOVA), and the multiple Range test were done using the Statgraphic Centurion XVII (Stat Point Inc., Warrenton, Virginia, USA). Differences were tested depending on cultivar, year and term of harvest, and interactions by LSD test with significance: non-significant (NS) or significant at *p* ≤ 0.05 (*), *p* ≤ 0.01 (**), or *p* ≤ 0.001 (***), respectively.

## 3. Results and Discussion

### 3.1. Yield Per Hectare

The total yield of petioles was estimated from data of six harvests per every tested cultivar grown at a density of 11,000 pl/ha. According to Table 5, the highest average values reached ‘Red Champagne’ (45.58 t/ha) and the lowest ‘Canadian Red’ (28.77 t/ha). As shown in Figure 1, the influence of the cultivar effect was significant (*p* < 0.001), as well as the year influence (*p* < 0.0025), when the average yield was higher in 2019.

The yield of the rhubarb cultivars is influenced both by the cultivar and the density according to Cojocaru et al. [14], where the total yield varied between 27.20 t/ha in the case of the ‘Glanskin’s perpetual’ cultivar, at a reduced density, up to 39.02 t/ ha in the case of the ‘Moldova Local population’ cultivar, at a high density. A study by Stoleru et al. [9] on the establishment of the rhubarb crop highlighted the fact that in the first year after a crop was established, there were obtained yields of 0.7 kg/pl. at a density of 8000 pl/ha. In the second and third year of a plantation, intensive plant growth was observed according to Salata and Kozak [22]. Our plants were harvested in the second and third years after the planting. The results correspond with the study by Stoleru et al. [9] who reported that, at a density of 13,300 pl/ha, the Victoria cultivar obtained an overall yield of 46.64 t/ha as compared with at a density of 8000 pl/ha, where the obtained yield was 21.72 t/ha. Lepse [23] observed no differences in total yield and yield quality in the second year of a plantation in an experimental study on rhubarb propagation by different methods. Very good commercial yields for rhubarb are approximately 1.5–3 kg of petioles per plant according to Welbaum [2], which corresponds with our results as shown in Figure 2. The average yields per plant varied between 2489.3 (‘Victoria’) and 3933.4 g/plant (‘Red Champagne’).

### 3.2. Juice Yield

As shown in Table 6, the juice yield significantly (*p* < 0.05) differed among the tested rhubarb cultivars. The average juice yield of the rhubarb cultivars ranged from 75% (‘Poncho’, HTA) to 85% (‘Red Champagne’, HTB). Generally, ‘Red Champagne’ was characterized by the highest juice yield potential for both harvests, as it significantly reached the highest values as compared with other tested cultivars. The statistical analysis of all tested results (Figure 3) showed significant differences (*p* < 0.001) in the case of morphological variability and influence of tested years on rhubarb juice yield. The term of harvests has not been shown to be significantly important to juice yield. The yield and composition of petioles are clearly influenced by the technology applied to a crop, especially the cultivar, the planting distance, and the nutritional regimen [14]. There were no data about juice yield of rhubarb found in the available literature. A comparison to similar celery petioles showed that high yield of celery juice was easy with a household juicer according to Donaldson, 2020 [24]. The juice yield and enzyme activity were tested in carrot, apples, spinach, and celery. The petiole juice yield of celery varied between 67.3% and 87.5% depending on juicer type. The yield of pressing ranged from 61.9% to 71.6% in the case of three apple cultivars according to Wilczyński et al. [11]. In addition, the dry matter of roots, in the case of carrot, were estimated to be from 41.1% to 65.2% depending on the juicer type [24].

### 3.3. Sugar Content

The average fructose content ranged from 33.93 (‘Valentine’, HTA) to 37.93 g/L (‘Valentine’, HTB), but cultivars significantly differed (*p* < 0.05) in fructose content only in the later harvest term (HTB), as shown in Table 7. When comparing all data from the cultivar variability point of view, the differences had not been proved (Figure 4). Fructose is highly represented in rhubarb. Its percentage representation is from 67% to 78% (as compared with total sugar content) according to the average from all results (Figure 5); therefore, it has high sensorial value in juice production. On the one hand, a statistically significant effect (*p* < 0.05) of the experimental year on fructose content was not found among rhubarb cultivars.

Values with different letters in columns are significantly different at *p* < 0.05 by LSD test in ANOVA (Statgraphic XVII). Abbreviations: PON, ‘Poncho’; CRE, ‘Canadian Red’; VAL, ‘Valentine’; RCH, ‘Red Champagne’; VIC, ‘Victoria’.

On the other hand, the term of harvest had a significant effect (*p* < 0.001) on the fructose content (Figure 4), which was higher in the second harvest.

According to Table 7, the glucose content in the tested cultivars varied from 6.26 (‘Victoria’, HTA) to 9.97 g/L (‘Red Champagne’, HTB), which was 14–16% of the total sugar content of the evaluated rhubarb cultivars (Figure 5). The obtained results confirmed the statistically significant impact of cultivars on the glucose content of rhubarb (Table 7). The higher values were found in the HTB and, like the case of fructose, those differences were proven at the *p* < 0.001 level (Figure 6). The impact of the year on glucose content was significantly important when following all data (for both harvests and all cultivars).

The total soluble solids, defined by Hegedűsová et al. [25] as additive quantity that expresses the content of dissolved substances, mainly sugars, in vegetable or fruit extracts, were also tested and are summarized in Table 8. The values for total soluble solids ranged from 3.54 (‘Valentine’, HTA) to 4.37 °BRIX (‘Red Champagne’, HTB). Significant differences (*p* < 0.05) were found between the tested cultivars and the values were higher in the second harvest (Table 8). When comparing all data in Figure 7, the significant effect (*p* < 0.001) of the cultivar, term of the harvest, and experimental year on the mean total soluble solids values of all rhubarb cultivars were found. The higher values in the late term of harvest could relate to the total soluble solid content increasing during ripening. This parameter is a very practical index of internal fruit quality and an accurate criterion for the decision to harvest in the field [26]. The mean value 3.8 °Brix was observed, with a range from 2.2 to 6.1° according to Pantoja and Kuhl [15], where they evaluated fifteen morphological characteristics to differentiate rhubarb cultivars.

The total sugar content varied between 44.19 (‘Valentine’, HTA) and 54.96 g/L (‘Red Champagne’, HTB) and according to Table 8 the differences between the tested cultivars were significant (*p* < 0.05). The values were higher at the later harvest (HTB), in which differences between the two harvests were also significantly proven at *p* < 0.001 (Figure 8). The year impact also significantly (*p* < 0.01) influenced the total sugar content parameter when comparing all cultivars and both harvests. The significant differences (*p* < 0.001) in terms of harvest influence on tested sugars as well as in the case of agrotechnical and soil/climate conditions through the tested years are in accordance with the study by Kalisz et al. [6] where the chemical composition of organic acids, minerals, carbohydrates, proteins, and vitamins highly depended on the cultivation and harvesting period.

### 3.4. Organic Acids and pH

Malic acid is the primary acid in rhubarb. Its value ranged from 14.06 (‘Red Champagne’, HTA) to 21.03 g/L (‘Valentine’, HTB). ‘Valentine’ reached the highest values for both harvests and the morphological variability was statistically improved at *p* < 0.05 (Table 9) in the frame of each harvest as well as in the statistical analyses of all data at *p* < 0.001 (Figure 9). Any significant differences were found in terms of harvest and the year influence on malic acid content (Figure 9). A similar situation was observed in the case of total acid content. In Table 9, cultivar variability on total acid content was significantly proven, where total acid content values were the highest in the case of the cultivar ‘Valentine’ for both harvests (21.88 g/L for HTA, and 22.95 g/L for HTB). The lowest value was reached by ‘Canadian Red’ at the second harvest (15.29 g/L) followed by ‘Red Champagne’ with a value of 15.45 g/L at the first harvest. As shown in Figure 10, cultivar variability had a significant impact (*p* < 0.001) on total acids content, as well as the year impact (*p* < 0.01). The term of harvest was not significantly important (*p* = 0.8747).

Organic acids play a biochemical role in maintaining the nutritional value and the quality of a vegetable species, and therefore they are among the frequently quantified compounds [27]. According to Welbaum [2], rhubarb leaves contain oxalic acid and should not be eaten. The petiole juice has a sharp acidic flavor. The oxalic acid in the petioles is much lower, the proportion of oxalic acid is about 10% of the total 2–2.5% acidity, which is dominated by nontoxic malic acid [7]. This means that the stalks are not hazardous to eat but have a very tart taste [2]. Malic acid is the predominant one, and citric and oxalic acids are also present in a smaller quantity in rhubarb plants according to Will and Dietrich [28]. The share of malic acid in total acid content varied between 89% and 92% in the evaluated rhubarb cultivars, as shown in Figure 5. Among the organic acids, the highest content was found in the case of malic acid, with average values of 679 ± 2.88 mg.100 g^−1^ fresh weight, according to Stoleru et al. [9]. Our results showed higher values because, in early petiole harvest, malic acid content is higher and decreases with vegetation period advancement in which the malic acid is naturally decomposed. According to the same study, the total acid content was 2239 mg.100 g^−1^ fresh weight. The chemical composition in organic acids highly depends on the cultivation and harvesting period [6].

The flavor of fruits and vegetable can also be determined by the sugar/acid (S/A) ratio that, in our case, expresses the ratio between total sugars and total acids; when the ratio is higher, the juice is subjectively less acidic. As shown in Table 10, the lowest S/A ratio of 2.02 was detected in the ‘Valentine’ cultivar, which means that there are 2.02 parts of sugars per one part of acid and the ‘Valentine’ is sensorially sweeter as compared with the ‘Red Champagne’ with the highest value of S/A ratio (3.35). The year and term of harvest did not have a significant effect on the S/A ratio according to the statistical analyses (Figure 11). Although the SSC/TA ratio is currently used as a maturity index for some types of fruit, it has been recognized that this measurement does not always correlate well with the perception of sweetness or tartness in other fruits [29].

The pH of rhubarb juice varied between 2.77 (‘Poncho’, THA) and 3.25 (‘Canadian Red’, HTB), as shown in Table 10, which corresponded with Welbaum [2], where the pH of rhubarb juice was measured to be 3.2; therefore, large quantities of sugar are often used to prepare rhubarb dishes. The cultivar had a significant influence (*p* < 0.001) on pH, as shown in Figure 12, while the year and term of harvest did not have a significant influence.

## 4. Conclusions

Since fresh rhubarb is considered to be not tasty, i.e., it is very sour and slightly sweet, the ideal solution for using its concentrated nutrients and antioxidants is to use it as a juice, or as an additive to mixed juices. Among the given cultivars, the highest yield was achieved by the cultivar ‘Red Champagne’ (45.58 t/ha), with the highest juice yield potential (85%), which is also interesting from the tested parameters point of view. It reached the highest values of glucose (9.97 g/L), total soluble solids (4.37 g/L), and total sugars (54.96 g/L). In contrast, ‘Valentine’ had glucose (6.50 g/L) and total sugar content (44.19 g/L) at the lowest level with the highest total acids (22.95 g/L). The juice yield of the plant was tested as an important parameter in the case of practical uses and its value varied in the following order: ‘Poncho’ < ‘Canadian Red’ < ‘Valentine’ < ‘Victoria’ < ‘Red Champagne’. The quality of the juice was significantly affected by the harvest term, where in the later term (HTB) the values were higher in the case of sugars (total sugars, TSS, fructose, and glucose). The acids were not changed by the term of harvest. In the case of some parameters, the significant influence of the year was also amplified, as there were significant differences in the agroclimatic characteristics tested for the two years. Since studying the sensory analysis of rhubarb juice is a new scientific issue, the results can serve as a basis for widespread use of this health-promoting crop, especially in the context of gastronomic use. We recommend adding rhubarb juice to vitalize, for example, apples harvested too late, where the share of malic acid is too low and achieves a balanced ratio between sugars and acids. The results show that rhubarb juice is a very suitable fortifier for such fruits, especially apples, which are unsaleable after long-term storage and juices made from such apples are often faint because malic acid has been degraded during the storage process. By enriching such juices with a rhubarb component, we achieve an optimal level of acidity in relation to the content of total sugars, which creates a very refreshing drink.

## Figures and Tables

**Figure 1 plants-10-01244-f001:**
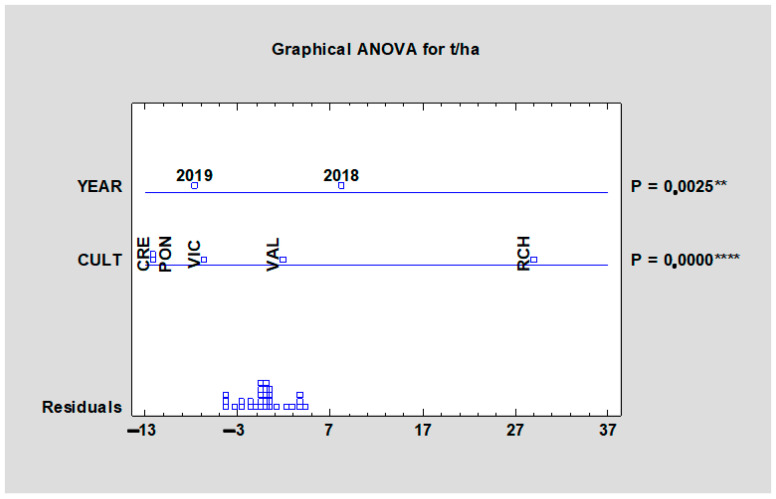
Effect of the experimental year and cultivar on the mean juice yield of all rhubarb cultivars. Abbreviations: PON, ‘Poncho’; CRE, ‘Canadian Red’; VAL, ‘Valentine’; RCH, ‘Red Champagne’; VIC, ‘Victoria’. LSD test with significance: significant at *p* ≤ 0.01 (**), or *p* = 0.0000 (****).

**Figure 2 plants-10-01244-f002:**
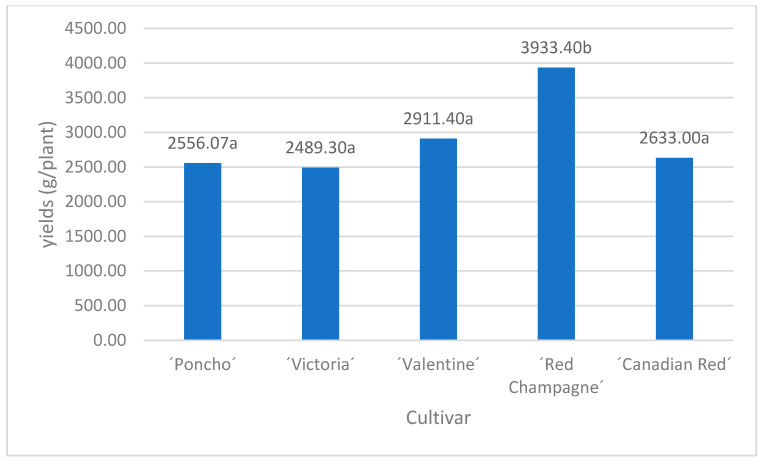
Average petiole yield for tested rhubarb cultivars. Values with different letters are significantly different at *p* < 0.05 by LSD test in ANOVA (Statgraphic XVII).

**Figure 3 plants-10-01244-f003:**
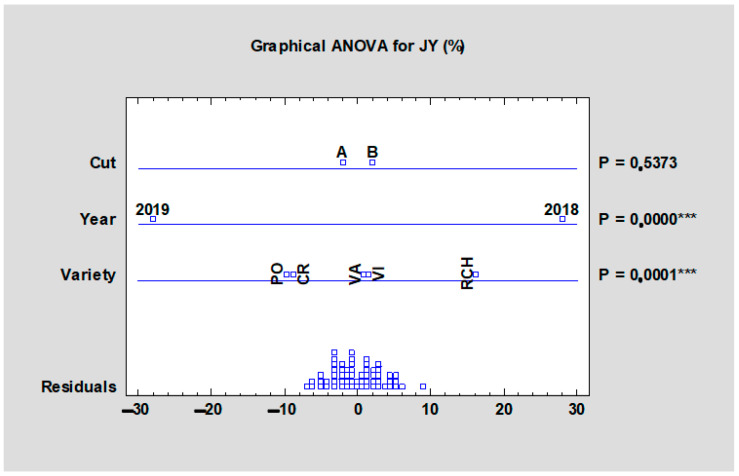
Effect of the experimental year, cultivar, and term of the harvest on the mean juice yield of all rhubarb cultivars. Abbreviations: PON, ‘Poncho’; CRE, ‘Canadian Red’; VAL, ‘Valentine’; RCH, ‘Red Champagne’; VIC, ‘Victoria’. LSD test with significance: *p* ≤ 0.001 (***).

**Figure 4 plants-10-01244-f004:**
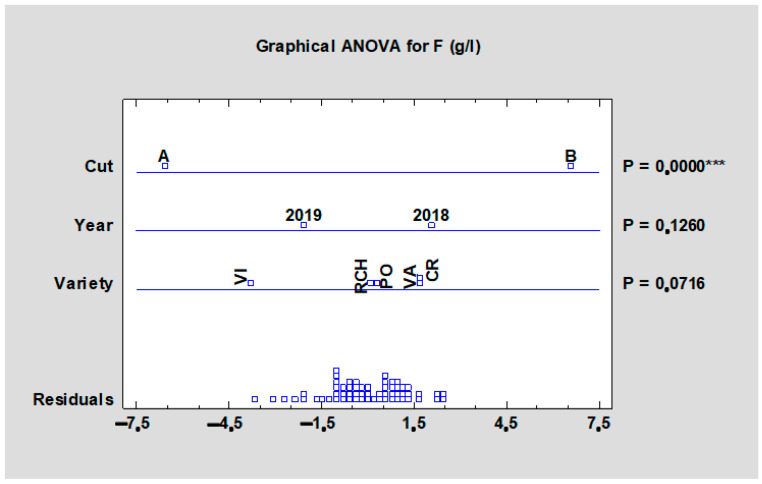
Effect of the experimental year, cultivar, and term of the harvest on the mean fructose content of all rhubarb cultivars. Abbreviations: PON, ‘Poncho’; CRE, ‘Canadian Red’; VAL, ‘Valentine’; RCH, ‘Red Champagne’; VIC, ‘Victoria’. LSD test with significance *p* ≤ 0.001 (***).

**Figure 5 plants-10-01244-f005:**
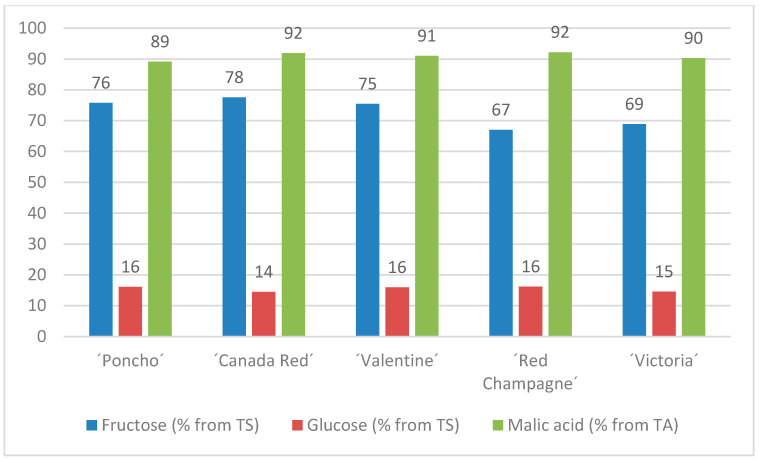
Share of fructose, glucose, and malic acid in total sugar and total acid content in the evaluated rhubarb cultivars. TS, total sugars and TA, total acid. Values are average from both harvests and years 2018–2019.

**Figure 6 plants-10-01244-f006:**
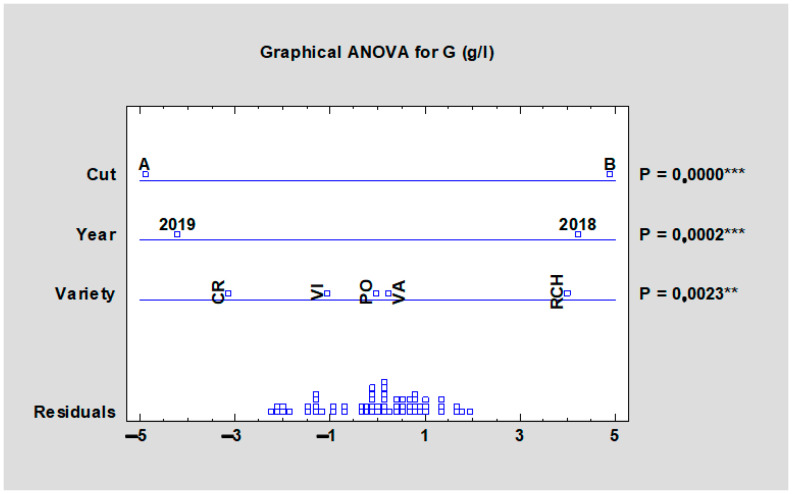
Effect of the experimental year, cultivar, and term of the harvest on the mean glucose content of all rhubarb cultivars. Abbreviations: PON, ‘Poncho’; CRE, ‘Canadian Red’; VAL, ‘Valentine’; RCH, ‘Red Champagne’; VIC, ‘Victoria’. LSD test with significance: *p* ≤ 0.01 (**), or *p* ≤ 0.001 (***).

**Figure 7 plants-10-01244-f007:**
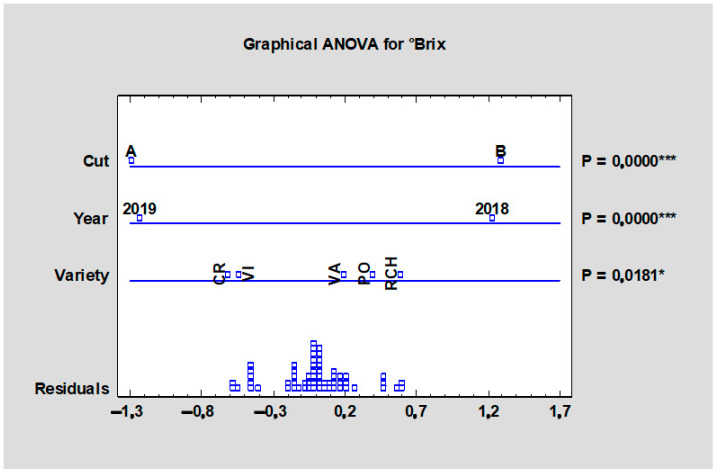
Effect of the experimental year, cultivar, and term of the harvest on the mean total soluble solid content of all rhubarb cultivars. Abbreviations: PON, ‘Poncho’; CRE, ‘Canadian Red’; VAL, ‘Valentine’; RCH, ‘Red Champagne’; VIC, ‘Victoria’. LSD test with significance: significant at *p* ≤ 0.05 (*) or *p* ≤ 0.001 (***).

**Figure 8 plants-10-01244-f008:**
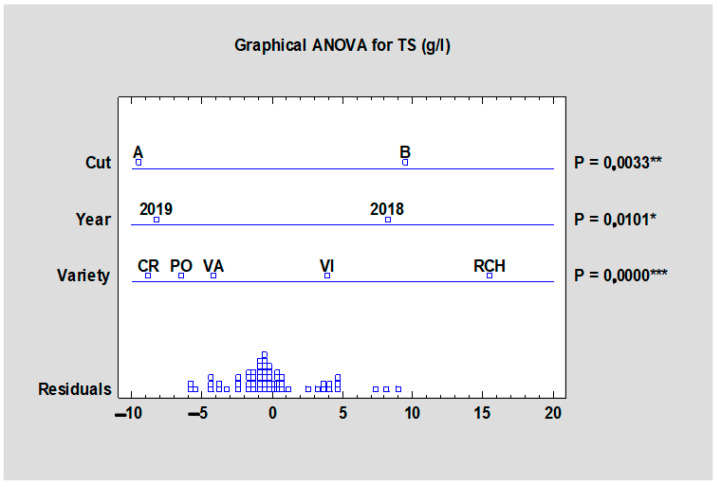
Effect of the experimental year, cultivar, and term of the harvest on the total sugar content of all rhubarb cultivars. Abbreviations: PON, ‘Poncho’; CRE, ‘Canadian Red’; VAL, ‘Valentine’; RCH, ‘Red Champagne’; VIC, ‘Victoria’. LSD test with significance: non-significant (NS) or significant at *p* ≤ 0.05 (*), *p* ≤ 0.01 (**), or *p* ≤ 0.001 (***).

**Figure 9 plants-10-01244-f009:**
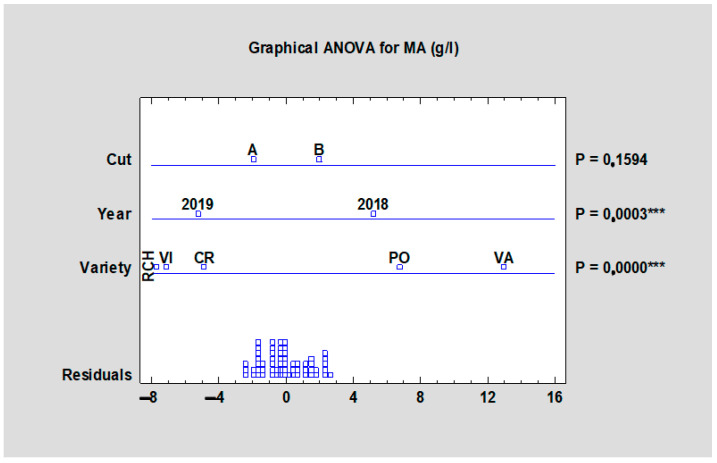
Effect of the experimental year, cultivar, and term of the harvest on the malic acid content in rhubarb cultivars. Abbreviations: PON, ‘Poncho’; CRE, ‘Canadian Red’; VAL, ‘Valentine’; RCH, ‘Red Champagne’; VIC, ‘Victoria’. LSD test with significance: *p* ≤ 0.001 (***).

**Figure 10 plants-10-01244-f010:**
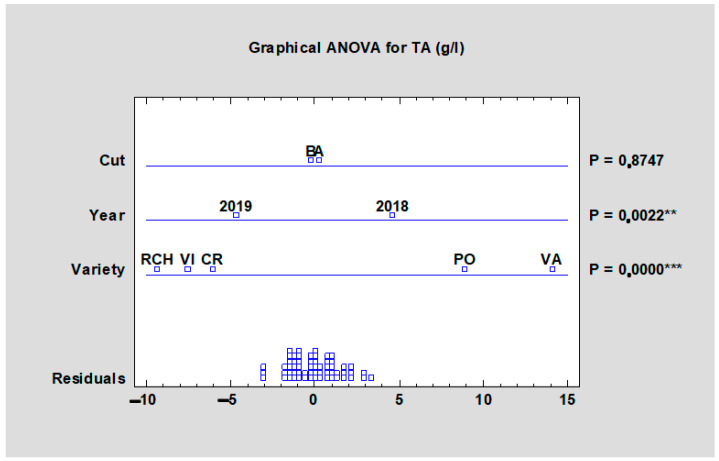
Effect of the experimental year, cultivar, and term of the harvest on the total acid content in rhubarb cultivars. Abbreviations: PON, ‘Poncho’; CRE, ‘Canadian Red’; VAL, ‘Valentine’; RCH, ‘Red Champagne’; VIC, ‘Victoria’. LSD test with significance: *p* ≤ 0.01 (**) or *p* ≤ 0.001 (***).

**Figure 11 plants-10-01244-f011:**
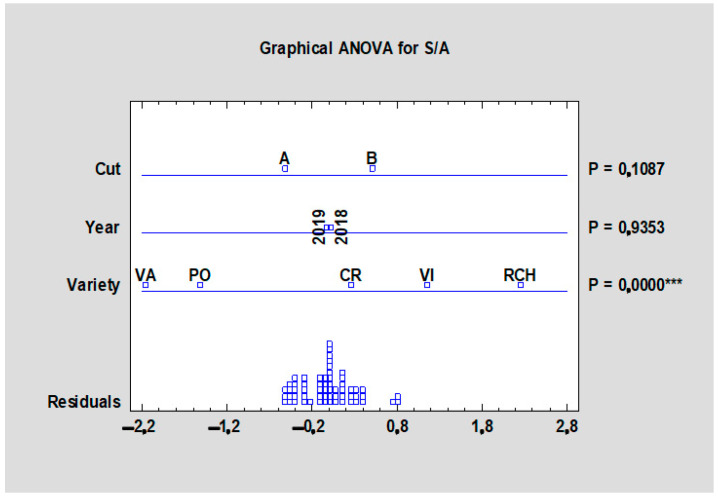
Effect of the experimental year, cultivar, and term of the harvest on the sugar/acid ratio in the rhubarb cultivars. Abbreviations: PON, ‘Poncho’; CRE, ‘Canadian Red’; VAL, ‘Valentine’; RCH, ‘Red Champagne’; VIC, ‘Victoria’. LSD test with significance: *p* ≤ 0.001 (***).

**Figure 12 plants-10-01244-f012:**
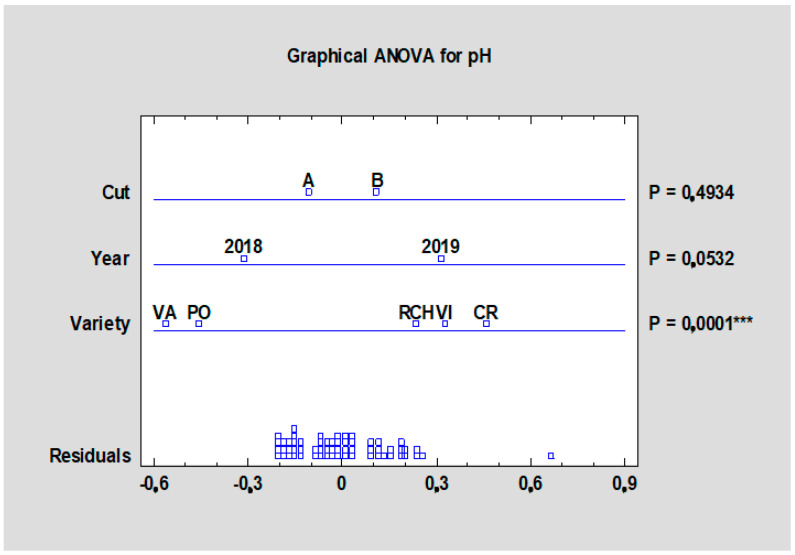
Effect of the experimental year, cultivar, and term of the harvest on the pH in the rhubarb cultivars. Abbreviations: PON, ‘Poncho’; CRE, ‘Canadian Red’; VAL, ‘Valentine’; RCH, ‘Red Champagne’; VIC, ‘Victoria’. LSD test with significance: *p* ≤ 0.001 (***).

**Table 1 plants-10-01244-t001:** Rhubarb cultivars’ characterization (morphological description according to UPOV (1999)) [18].

	Morphological Description of Petiole	Picture of the Petioles *
‘Poncho’	Semi-erect attitude, type of cross-section 1, green ground color of skin, entire distribution of skin superimposed color at base, absent distribution of skin superimposed color at middle, absent distribution of skin superimposed color just below leaf blade, present hairiness just below leaf blade, absent or very weak ribbing of dorsal side, green color of flesh	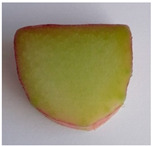
‘Victoria’	Erect attitude, type of cross-section 2, green ground color of skin, entire distribution of skin superimposed color at base, speckled distribution of skin superimposed color at middle, absent distribution of skin superimposed color just below leaf blade, present hairiness just below leaf blade, weak ribbing of dorsal side, green color of flesh	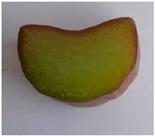
‘Valentine’	Erect attitude, type of cross-section 3, red ground color of skin, entire distribution of skin superimposed color at base, speckled distribution of skin superimposed color at middle, speckled distribution of skin superimposed color just below leaf blade, absent hairiness just below leaf blade, medium ribbing of dorsal side, pink color of flesh	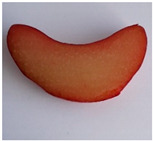
‘Red Champagne’	Erect attitude, type of cross-section 2, red ground color of skin, entire distribution of skin superimposed color at base, speckled distribution of skin superimposed color at middle, speckled distribution of skin superimposed color just below leaf blade, present hairiness just below leaf blade, weak ribbing of dorsal side, green color of flesh	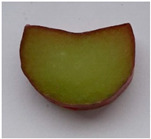
‘Canadian Red’	Semi-erect attitude, type of cross-section 7, red ground color of skin, entire distribution of skin superimposed color at base, entire distribution of skin superimposed color at middle, speckled distribution of skin superimposed color just below leaf blade, present hairiness just below leaf blade, strong ribbing of dorsal side, green color of flesh	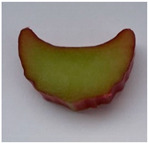

* Photo by Andrejiová.

**Table 2 plants-10-01244-t002:** Quantitative parameters of *Rheum rhabarbarum* L. petioles (*n* = 100).

Cultivar	Length (cm)	Width (cm)	Thickness (cm)	Weight (g)
‘Poncho’	29.42 ± 6.21 a	1.79 ± 0.33 a	1.16 ± 0.29 cd	49.35 ± 21.88 a
‘Victoria’	33.99 ± 6.39 b	1.78 ± 0.28 a	0.97 ± 0.19 a	54.38 ± 15.80 a
‘Valentine’	33.38 ± 6.64 b	2.31 ± 0.34 c	1.29 ± 0.30 d	77.61 ± 33.60 c
‘Red Champagne’	37.03 ± 7.14 c	1.96 ± 0.48 b	1.02 ± 0.22 ab	63.04 ± 25.58 b
‘Canadian Red’	29.55 ± 4.09 a	2.27 ± 0.30 c	1.07 ± 0.28 bc	62.47 ± 20.43 b

Values with different letters are significantly different at *p* < 0.05 by LSD test in ANOVA (Statgraphic XVII)

**Table 3 plants-10-01244-t003:** Evaluation of months according to air temperature climate normals 1961–1990.

Month	Normal 1961–1990 (°C)	T (°C) 2018	Characteristic—2018	T (°C) 2019	Characteristic—2019
III	5.0	3.4	cold	8.1	very hot
IV	10.4	15.4	extremely hot	9.4	normal
V	15.1	18.8	very hot	9.3	very cold
VI	18.0	20.7	hot	18.7	normal
VII	19.8	21.7	hot	21.9	hot

**Table 4 plants-10-01244-t004:** Evaluation of months according to annual precipitation climate normals 1961–1990.

Month	Normal 1961–1990 (mm)	PRC (mm) 2018	Characteristic—2018	PRC (mm) 2019	Characteristic—2019
III	30	36	normal	16	very dry
IV	39	16	very dry	21	dry
V	58	29	very dry	135	extremely wet
VI	66	44	dry	29	very dry
VII	52	13	very dry	21	very dry

**Table 5 plants-10-01244-t005:** The yields of rhubarb petioles.

Cultivar	Yield (t/ha)		
	2018	2019	2018–2019
PON	30.48 ± 4.22 a	28.40 ± 1.10 ab	29.44 ± 1.47 a
CRE	29.87 ± 1.12 a	27.66 ± 3.20 a	28.77 ± 1.56 a
VAL	36.61 ± 0.36 b	32.35 ± 3.32 b	34.48 ± 3.01 b
RCH	47.46 ± 3.60 c	43.70 ± 1.24 c	45.58 ± 2.65 c
VIC	32.70 ± 4.14 ab	29.26 ± 2.36 ab	30.98 ± 2.43 ab

Values with different letters are significantly different at *p* < 0.05 by LSD test in ANOVA (Statgraphic XVII). Abbreviations: PON, ‘Poncho’; CRE, ‘Canadian Red’; VAL, ‘Valentine’; RCH, ‘Red Champagne’; VIC, ‘Victoria’.

**Table 6 plants-10-01244-t006:** Influence of cultivar and term of harvest on juice yield of rhubarb petioles.

Cultivar	2018 (HTA)	2019 (HTA)	2018–2019 (HTA)
	JY (%)	JY (%)	JY (%)
**PON**	77 ± 2 a	74 ± 3 abc	75 ± 1 a
**CRE**	79 ± 2 ab	80 ± 3 cd	79 ± 2 abc
**VAL**	88 ± 2 c	72 ± 2 ab	80 ± 3 abc
**RCH**	90 ± 3 c	79 ± 3 bcd	84 ± 2 c
**VIC**	87 ± 3 c	78 ± 2 bcd	82 ± 2 bc
	**2018 (HTB)**	**2019 (HTB)**	**2018–2019 (HTB)**
**PON**	84 ± 2 bc	76 ± 2 abc	80 ± 3 abc
**CRE**	83 ± 3 abc	70 ± 2 a	77 ± 2 ab
**VAL**	84 ± 3 bc	80 ± 2 cd	82 ± 2 bc
**RCH**	87 ± 1 c	84 ± 3 d	85 ± 1 c
**VIC**	84 ± 2 abc	75 ± 1 abc	79 ± 2 abc

Values with different letters are significantly different at *p* < 0.05 by LSD test in ANOVA (Statgraphic XVII). Abbreviations: PON, ‘Poncho’; CRE, ‘Canadian Red’; VAL, ‘Valentine’; RCH, ‘Red Champagne’; VIC, ‘Victoria’.

**Table 7 plants-10-01244-t007:** Effect of cultivar on fructose and glucose content in rhubarb juice.

Cultivar	2018 (term HTA)		2019 (HTA)		2018–2019 (HTA)	
	Fructose (g/L)	Glucose (g/L)	Fructose (g/L)	Glucose (g/L)	Fructose (g/L)	Glucose (g/L)
**PON**	35.40 ± 0.34 cd	7.88 ± 0.19 d	35.03 ± 0.27 c	7.77 ± 0.18 e	35.22 ± 0.26 abc	7.83 ± 0.08 cde
**CRE**	34.58 ± 0.20 b	7.47 ± 0.16 c	34.40 ± 0.36 bc	5.69 ± 0.21 c	34.49 ± 0.13 ab	6.58 ± 1.26 ab
**VAL**	33.35 ± 0.45 a	6.34 ± 0.28 a	34.50 ± 0.21 bc	6.65 ± 0.12 d	33.93 ± 0.82 ab	6.50 ± 0.22 ab
**RCH**	35.50 ± 0.80 cd	9.48 ± 0.08 f	34.05 ± 0.71 b	5.33 ± 0.14 b	34.78 ± 1.03 ab	7.40 ± 2.93 bcd
**VIC**	36.03 ± 0.43 d	7.78 ± 0.07 d	32.80 ± 0.53 a	4.74 ± 0.07 a	34.41 ± 2.28 a	6.26 ± 2.15 a
	**2018 (HTB)**		**2019 (HTB)**		**2018–2019 (HTB)**	
**PON**	33.58 ± 0.49 a	6.86 ± 0.10 b	38.18 ± 0.58 e	7.81 ± 0.15 e	35.88 ± 3.25 bc	7.34 ± 0.67 abcd
**CRE**	38.20 ± 0.41 ef	6.90 ± 0.21 b	36.63 ± 0.50 d	6.82 ± 0.15 d	37.42 ± 1.11 de	6.86 ± 0.05 abc
**VAL**	38.46 ± 0.30 f	8.71 ± 0.12 e	37.40 ± 0.27 de	8.89 ± 0.18 g	37.93 ± 0.75 e	8.80 ± 0.12 e
**RCH**	37.56 ± 0.09 e	11.74 ± 0.13 g	34.87 ± 1.00 bc	8.20 ± 0.21 f	36.21 ± 1.90 cd	9.97 ± 2.51 f
**VIC**	34.89 ± 0.44 bc	8.48 ± 0.13 e	33.99 ± 0.19 b	8.19 ± 0.10 f	34.44 ± 0.64 a	8.33 ± 0.21 de

Values with different letters in columns are significantly different at *p* < 0.05 by LSD test in ANOVA (Statgraphic XVII). Abbreviations: PON, ‘Poncho’; CRE, ‘Canadian Red’; VAL, ‘Valentine’; RCH, ‘Red Champagne’; VIC, ‘Victoria’.

**Table 8 plants-10-01244-t008:** Effect of cultivar on total soluble solid content and total sugar content in rhubarb juice.

Cultivar	2018 (HTA)		2019 (HTA)		2018–2019 (HTA)	
	TSS	Total sugar	TSS	Total sugar	TSS	Total sugar
	°BRIX	(g/L)	°BRIX	(g/L)	°BRIX	(g/L)
**PON**	3.98 ± 0.03 d	44.89 ± 0.40 c	4.22 ± 0.02 h	52.82 ± 0.80 f	4.10 ± 0.17 cd	48.86 ± 5.61 bc
**CRE**	3.74 ± 0.02 b	45.35 ± 0.14 c	3.44 ± 0.04 b	43.22 ± 0.11 a	3.59 ± 0.21 ab	44.29 ± 1.50 a
**VAL**	3.48 ± 0.03 a	43.32 ± 0.12 a	3.60 ± 0.02 c	45.06 ± 0.15 b	3.54 ± 0.08 a	44.19 ± 1.23 a
**RCH**	4.15 ± 0.02 f	52.59 ± 0.10 ef	3.25 ± 0.01 a	50.14 ± 0.37 e	3.70 ± 0.63 ab	51.36 ± 1.73 c
**VIC**	3.87 ± 0.04 c	50.69 ± 0.22 d	3.23 ± 0.01 a	47.90 ± 0.48 d	3.55 ± 0.45 a	49.30 ± 1.91 c
	**2018 (HTB)**		**2019 (HTB)**		**2018–2019 (HTB)**	
**PON**	3.76 ± 0.03 b	43.99 ± 0.16 b	3.95 ± 0.02 f	46.82 ± 0.11 c	3.86 ± 0.13 abc	45.40 ± 2.00 ab
**CRE**	4.04 ± 0.05 e	52.12 ± 0.57 e	3.60 ± 0.03 c	45.25 ± 0.19 b	3.82 ± 0.32 abc	48.69 ± 4.86 bc
**VAL**	4.50 ± 0.08 g	54.43 ± 0.59 h	4.12 ± 0.02 g	48.20 ± 0.23 d	4.31 ± 0.27 d	51.31 ± 4.41 c
**RCH**	4.86 ± 0.06 h	60.14 ± 0.43 i	3.88 ± 0.03 e	49.78 ± 0.26 e	4.37 ± 0.70 d	54.96 ± 7.33 d
**VIC**	4.03 ± 0.12 e	52.85 ± 0.33 g	3.76 ± 0.03 d	48.51 ± 0.43 d	3.90 ± 0.20 bc	50.68 ± 3.07 c

Values with different letters in columns are significantly different at *p* < 0.05 by LSD test in ANOVA (Statgraphic XVII). Abbreviations: PON, ‘Poncho’; CRE, ‘Canadian Red’; VAL, ‘Valentine’; RCH, ‘Red Champagne’; VIC, ‘Victoria’.

**Table 9 plants-10-01244-t009:** Effect of cultivar on malic acid and total acid content in rhubarb juice.

Cultivar	2018 (HTA)		2019 (HTA)		2018–2019 (HTA)	
	Malic Acid	Total Acids	Malic Acid	Total Acids	MALIC ACID	Total Acids
	(g/L)	(g/L)	(g/L)	(g/L)	(g/L)	(g/L)
**PON**	21.62 ± 0.16 h	24.73 ± 2.23 i	15.34 ± 0.59 d	17.43 ± 0.55 e	18.48 ± 4.45 c	21.08 ± 5.16 c
**CRE**	16.47 ± 1.06 e	18.49 ± 0.10 e	16.99 ± 1.04 g	18.42 ± 1.04 f	16.73 ± 0.37 b	18.46 ± 0.25 b
**VAL**	19.40 ± 2.03 f	21.92 ± 1.05 f	20.19 ± 2.06 j	21.84 ± 0.88 h	19.80 ± 0.56 cd	21.88 ± 0.46 cd
**RCH**	14.43 ± 1.07 a	15.17 ± 0.03 a	13.69 ± 0.16 c	15.72 ± 1.06 b	14.06 ± 0.52 a	15.45 ± 0.39 a
**VIC**	15.49 ± 0.94 c	16.14 ± 0.65 c	12.13 ± 0.17 a	15.89 ± 2.14 c	13.81 ± 2.38 a	16.01 ± 0.17 a
	**2018 (HTB)**		**2019 (HTB)**		**2018–2019 (HTB)**	
**PON**	19.79 ± 1.03 g	22.39 ± 1.06 g	18.03 ± 1.05 h	19.44 ± 2.07 g	18.91 ± 1.25 c	20.92 ± 2.08 c
**CRE**	15.08 ± 0.25 b	16.16 ± 1.03 c	13.44 ± 1.09 b	14.42 ± 0.22 a	14.26 ± 1.16 a	15.29 ± 1.23 a
**VAL**	22.63 ± 2.08 i	24.16 ± 2.13 h	19.43 ± 1.06 i	21.74 ± 0.83 h	21.03 ± 2.27 d	22.95 ± 1.71 d
**RCH**	14.97 ± 1.06 b	15.90 ± 1.09 b	15.81 ± 0.25 e	17.10 ± 1.05 d	15.39 ± 0.60 ab	16.50 ± 0.85 a
**VIC**	15.71 ± 1.12 d	16.75 ± 0.14 d	16.21 ± 0.39 f	17.06 ± 2.08 d	15.96 ± 0.35 b	16.90 ± 0.22 ab

Values with different letters in columns are significantly different at *p* < 0.05 by LSD test in ANOVA (Statgraphic XVII). Abbreviations: PON, ‘Poncho’; CRE, ‘Canadian Red’; VAL, ‘Valentine’; RCH, ‘Red Champagne’; VIC, ‘Victoria’.

**Table 10 plants-10-01244-t010:** Effect of cultivar on pH level and the S/A ratio in rhubarb juice.

Cultivar	2018 (HTA)		2019 (HTA)		2018–2019 (HTA)	
	pH	S/A Ratio	pH	S/A Ratio	pH	S/A Ratio
**PON**	2.60 ± 0.02 a	1.82 ± 0.03 a	2.95 ± 0.04 d	3.03 ± 0.05 g	2.77 ± 0.08 b	2.42 ± 0.86 b
**CRE**	2.86 ± 0.22 bc	2.45 ± 0.01 d	2.93 ± 0.12 d	2.35 ± 0.00 c	2.90 ± 0.24 b	2.40 ± 0.07 b
**VAL**	2.73 ± 0.13 ab	1.98 ± 0.00 b	2.76 ± 0.14 a	2.06 ± 0.00 a	2.74 ± 0.25 a	2.02 ± 0.06 a
**RCH**	2.99 ± 0.21 c	3.47 ± 0.01 g	3.18 ± 0.18 g	3.19 ± 0.03 i	3.09 ± 0.06 cd	3.33 ± 0.20 cd
**VIC**	3.00 ± 0.32 c	3.14 ± 0.00 e	3.31 ± 0.11 h	3.01 ± 0.05 g	3.16 ± 0.04 de	3.08 ± 0.09 cd
	**2018 (B)**		**2019 (B)**		**2018–2019 (B)**	
**PON**	2.66 ± 0.04 a	1.96 ± 0.01 b	3.08 ± 0.18 e	2.41 ± 0.00 d	2.87 ± 0.29 b	2.19 ± 0.31 ab
**CRE**	3.39 ± 0.28 d	3.23 ± 0.04 f	3.11 ± 0.16 f	3.14 ± 0.02 h	3.25 ± 0.20 e	3.18 ± 0.06 cd
**VAL**	2.88 ± 0.03 c	2.25 ± 0.03 c	2.80 ± 0.06 b	2.22 ± 0.01 b	2.84 ± 0.09 b	2.24 ± 0.03 ab
**RCH**	2.99 ± 0.02 c	3.78 ± 0.05 h	2.88 ± 0.09 c	2.91 ± 0.02 f	2.93 ± 0.06 bc	3.35 ± 0.62 d
**VIC**	2.93 ± 0.01 c	3.16 ± 0.04 e	2.90 ± 0.06 c	2.84 ± 0.03 e	2.92 ± 0.11 b	3.00 ± 0.22 c

Values with different letters in columns are significantly different at P < 0.05 by LSD test in ANOVA (Statgraphic XVII). Abbreviations: PON, ‘Poncho’; CRE, ‘Canadian Red’; VAL, ‘Valentine’; RCH, ‘Red Champagne’; VIC, ‘Victoria’.

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
