# Peer review of "The Effect of the Cultivar and Harvest Term on the Yield and Nutritional Value of Rhubarb Juice"

_plants, 2021, doi:10.3390/plants10061244_

Round 1

Reviewer 1 Report

The manuscript entitled "" is an interesting one, but in order to be published it needs a major revision.
Abstract. We cannot say that in literature there are no studies on rubraba because it exists. The event we can mention that they are few.
Introduction. I suggest details on the characterization of rhubarb in the literature, not just listed.
Table 2 - statistical analysis is missing
I do not understand why the temperatures and precipitation from 1961-1990 were taken into analysis, since the study was conducted in 2018-2019.
Were the temperature and precipitation monitored by the authors or were they taken from specialized institutions? Why were rainfall and temperatures compared to 1961-1990?
Table 5 lacks the statistical analysis for the years 2018 and 2019, it appears only for the average 2018-2019
I suggest adding statistics in Figure 2
Table 6 lacks the statistical analysis for the years 2018 and 2019, it appears only for the average 2018-2019
Table 7, 8, 9, 10 lacks statistical analysis for the years 2018 and 2019, appears only for the average 2018-2019
Why were only fructose and glucose analyzed?
Comments on the total number of acids are missing
Description of methods for fructose, glucose and malic acid, pH, ° Brix, Total sugar is missing
Why has the total profile of organic acids not been studied?
The results are not correlated with meteorological data
I suggest improving the comments
I suggest reformulating the conclusions regarding the analyzes performed
The bibliography must be redone according to the requirements of the journal

Author Response

Dear Reviewer, thank you for the submitted suggestions, comments. We tried to accept all the proposals. Along with the answers we are explaining all the changes we have done. All changes made are marked in red. We would like to thank you once again for your consideration of our work, for pointing out that mistakes and inviting us to submit.

Reviewer 2 Report

The manuscript titled “Effect of the cultivar and harvest term on the yield and nutrition level of rhubarb juice” presents the characterization of a new fruit juice with potential bioactive properties. Although it does not tackle the bioactive aspect of the work, the manuscript brings information that can be interesting from an industrial point of view. However, some improvement is necessary, as outlined below. 

Abstract:

Line 16: “were determined” is the correct form.

Line 17: The authors can remove equipment information from the abstract.

Line 18: replace “moved from… to…” with “varied between… and…”

Line 20: include the numerical results for glucose, soluble solids, and sugars for the Red Champagne sample.

Introduction:

Lines 36-37: I believe this information should come before mentioning the health effects, as the latter is a consequence of the former, not the other way around.

Lines 48-49: this recommendation is based on what?

Line 50: reducing the risk of serious illness is a way to promote good health. The authors wrote that if those were two different things and that’s not the case.

Line 67: Does the use of fruits to produce juice really characterizes a new perspective? I understand where the authors are coming from here, but I think this statement should be re-written in order to better express the true meaning of what the authors are trying to say.
Comment: Why haven’t the authors also explored the bioactive potential of the juice? This was one of the main reasons the authors gave in the introduction for the consumption of this product, so I don’t understand why this aspect was left out of the study.

Material and Methods:

Table 3: The description of the weather characteristic does not seem to match the temperatures. How come 15 degrees be described as extremely hot while 21 degrees is described as hot? This is a repeating issue in tables 3 and 4.

Line 124: were determined.

Item 2.6: method reference?

Results and Discussion:

Line 165: Wouldn’t it be Table 6 instead?

Line 251: replace “most represented” with “primary.”

Conclusion:

Future studies suggestions?

Author Response

(The authors gave the same response as above.)

Reviewer 3 Report

In this manuscript (ID:plants-1249348), authors statistically analyzed effects of cultivar, year, and harvest term on the petiole and juice yield of rhubarb, and on contents of ingredients (fructose, glucose, total sugars, malic acid, total acids, and total soluble solid) and pH of rhubarb juice. Object of this study and presentation of data are clear. However, the method for determination of contents of ingredients isn’t seemed to be suitable. Authors used FT-IR spectrophotometer for wine and must, to obtain data of rhubarb juice, but there is no description on the calibration of actual measurement of rhubarb juice ingredients, e.g. fructose, glucose, malic acid, and so on. Description method of numerical value is ununified in the manuscript.

Each comment is described below.

  1. Lines 39-41: There is no description on the sentence in the reference number 2.
  2. Line 58: There is duplication of “it is”.
  3. Lines 66-67: If the reason of the selection of cultivars used in this study is low oxalate, the content of oxalate in each sample should be measured or described in the manuscript.
  4. Line 95: Isn’t it density of 11,000 pl/ha? There are similar values of density in this text.
  5. Line 108: Is it really 100 m long pieces?
  6. Lines 320-321: Contents of malic acid and total acids in juice of ‘Canadian Red’ are significantly larger in HTA than those in HTB, according to Table 9.
  7. Table 10: Units of pH and S/A ratio are wrong.
  8. References: There are many omissions, e.g. pages, journal name, year of publication.

For example, in reference number 17, journal name and pages are omissions, and in reference number 12 and 20, journal name, volume number, pages and year of publication are omissions.

Author Response

(The authors gave the same response as above.)

Round 2

Reviewer 1 Report

The authors responded to all my comments.

Author Response

Please read the appendix.

Reviewer 3 Report

Dear Authors,

Thank you for revision of the manuscript.

In the revised manuscript, It is easy to understand the mechanism of the FT-IR spectrophotometer, with reference of the analyzer. I had gotten a manual of this analyzer from web site, and I know that many of parameters (fructose, glucose, total sugar, malic acid, total acid and Brix) used for rhubarb juice in this manuscript are out of concentration ranges of the initial calibration of wine and must, as you respond to my comment. So I think that it is necessary to prepare a user calibration for adequate concentrations of fructose, glucose, and so on, or each sample (rhubarb juice) should be diluted or concentrated before analysis, as each parameter's fall within concentration range.

I agree with the revised manuscript except for the above point.

Author Response

Please read the appendix. 

This manuscript is a resubmission of an earlier submission. The following is a list of the peer review reports and author responses from that submission.